# Establishment of an In Vitro Model of Persistent Chicken Anemia Virus Infection

**DOI:** 10.3390/pathogens9100842

**Published:** 2020-10-15

**Authors:** Hieu Van Dong, Giang Thi Huong Tran, Dai Quang Trinh, Yohei Takeda, Haruko Ogawa, Kunitoshi Imai

**Affiliations:** 1United Graduate School of Veterinary Sciences, Gifu University, 1-1 Yanagido, Gifu City 501-1193, Japan; dvhieuvet1984@gmail.com (H.V.D.); giangtranvet@gmail.com (G.T.H.T.); 2Department of Veterinary Medicine, Obihiro University of Agriculture and Veterinary Medicine, 2-11 Inada, Obihiro, Hokkaido 080-8555, Japan; imaiku@obihiro.ac.jp; 3Faculty of Veterinary Medicine, Vietnam National University of Agriculture, Trau Quy Town, Gia Lam, Hanoi 131000, Vietnam; 4Central Veterinary Medicine JSC No. 5, Ha Binh Phuong Industrial Zone, Thuong Tin, Hanoi 131000, Vietnam; trinhdai2004@yahoo.com; 5Research Center for Global Agromedicine, Obihiro University of Agriculture and Veterinary Medicine, 2-11 Inada, Obihiro, Hokkaido 080-8555, Japan; ytakeda@obihiro.ac.jp

**Keywords:** chicken anemia virus, in vitro model, neutralizing antibody, persistent infection

## Abstract

Persistent infection of chicken anemia virus (CAV) in chickens has been suspected to result in immunosuppression and exogenous virus contamination within vaccine production. However, no direct evidence for persistent CAV infection has thus far been obtained. In this study, we aimed to establish an in vitro model of persistent CAV infection. CAV-infected MDCC-MSB1 (MSB1) cells, a Marek’s disease virus-transformed continuous cell line, were cultured in the presence of both CAV and CAV neutralizing antibody (NA). Cell viability, expression of viral antigens, viral DNA, and recovery of CAV were examined by acridine orange/propidium iodide staining, immunofluorescence measurement, real-time PCR, and viral isolation, respectively. The results indicated that CAV was maintained and possibly replicated in CAV-infected cells cultured in the presence of NA, without affecting host cell viability. It was also shown that persistently infectious CAV induced cell death again after removing NA. The persistent infection of CAV in MSB1 cells was not related to viral gene mutation. In summary, we have herein established a novel model of persistent CAV infection in MSB1 cells cultured in the presence of NA.

## 1. Introduction

Chicken anemia virus (CAV), a member of the genus *Gyrovirus* in the family Anelloviridae, is characterized as a non-enveloped, spherical small virus [1]. The viral genome consists of circular, negative-sense single-stranded DNA with three open reading frames (ORFs)—1, 2, and 3. These ORFs encode three viral proteins (VPs), the major capsid structural protein VP1, scaffolding VP2, and strong inducer of apoptosis VP3 (apoptin), respectively [1,2]. The VP1 protein is known to induce the production of neutralizing antibody (NA) in the host and is produced within the early phase of infection at 12 h post-infection (hpi) in Marek’s disease virus-transformed continuous cell line MDCC-MSB1 (MSB1) cells [3]. VP2 serves as a scaffolding protein for VP1 and plays a critical role in phosphatase activity. Apoptin contributes to the induction of apoptosis both in vitro and in vivo [4,5]. Of note, the interaction between VP2 and apoptin influences the downregulation of apoptosis in vitro [6].

Following the identification of the virus in 1979 in Japan by Yuasa and colleagues [7], CAV has gained worldwide attention as an infectious agent affecting the chicken production industry [8]. It has been shown that the virus can be spread via both horizontal and vertical transmissions in chickens [9,10,11]. Horizontal transmission occurs after the disappearance of maternal antibodies, resulting in sub-clinical disease in chickens older than two weeks of age, with affected chickens suffering from immunosuppression [12,13,14]. Vertical transmission in antibody-negative hens (i.e., breeders) to their progeny plays a critical role within the clinical disease observed in young chickens, resulting in anemia, depression, increased mortality, and economic losses in the field [15,16,17].

Specific-pathogen-free (SPF) chickens and their eggs are essential for human and animal vaccine production. It is worth noting that CAV infection has been reported in commercial SPF chicken flocks [18,19]. Seroconversion resulting from CAV infection in SPF chickens is reported even in renewed chicken flocks and under strict hygiene operation [20,21]. Affected chicken eggs are unsuitable for use in vaccine production. It has been a challenge to eradicate CAV from infected flocks following the introduction of the virus. Brentano et al. [22] reported that CAV DNA was detectable not only in the reproductive organs of hens with high titers of NA but also in their embryonated eggs. These findings raised the possibility of the existence of persistent CAV infection. However, no hard evidence for CAV persistence within the host has thus far been obtained. In this study, we aimed to establish an in vitro model for future studies of CAV persistent infection.

## 2. Results

### 2.1. Cell Viability

During the time course, percent of cell viability gradually decreased at 48 (60.83 ± 1.4), 72 (53.17 ± 2.27), and 96 (32.17 ± 2.23) hpi for CAV-infected cells cultured in the absence of NA (i.e., CAV-infected), which was significantly lower than that of CAV-infected cells cultured in the presence of NA (CAV-infected + NA) (*p* < 0.05) at each time point (Figure 1A). Thus, CAV infection did result in cell death, which was suppressed in the presence of NA.

In the passage study, cell viability monitored in the CAV-infected + NA cells at each 48 h period was significantly higher (*p* < 0.01) than that of the CAV-infected cells after the first passage (P1). Cell viability dropped gradually to 23.15 ± 3.32 percent at P3 in the CAV-infected cells. In contrast, infected cells were still viable in the CAV-infected + NA cultures until P14 (Figure 1B). In order to demonstrate whether non-neutralizing antibody affects CAV-infected cells or not, an additional study on CAV-infected cells cultured in the presence of SPF chicken serum (i.e., CAV-infected + SPFs) was performed in the passage study. The results indicated that the cell viability of CAV-infected + SPFs cells was significantly lower than that of CAV-infected + NA after the first (*p* < 0.05), second (*p* < 0.01), and third passage (*p* < 0.001). No significant differences (*p* > 0.05) were observed in cell viability between the CAV-infected cells and CAV-infected + SPFs cells during P1 to P3 (Figure 1C).

### 2.2. Detection of CAV Antigens

CAV antigens in the infected cells were detectable by an indirect fluorescent antibody test (IFAT) using antiserum to the CAV/A2/76 strain (chicken anti-A2/76). The granular shapes observed were regarded as CAV antigens in both CAV-infected and the CAV-infected + NA cells at 24, 48, 72, and 96 hpi (Figure 2). CAV antigens were also confirmed in CAV-infected + NA cells at the fifth passage (P5), P10, and P14. By contrast, no fluorescence signal was observed in non-infected cells at the same passages (Figure 3A). VP1, VP2, and VP3 antigens were detected in CAV-infected + NA cells at P14 (Figure 3B,C).

### 2.3. Detection of the CAV Genome within Infected Cells and Cell Supernatants

Real-time PCR was conducted to detect CAV DNA, and results were calculated as the log_10_ number of copies per one million cells for infected cells and log_10_ number of copies/mL within cell supernatants (sups). There was no significant difference in the amount of viral DNA between the CAV-infected (9.45 ± 0.09 log_10_ copies/10^6^ cells) and CAV-infected + NA (9.29 ± 0.02 log_10_ copies/10^6^ cells) cells at 24 hpi. The results were similar at 48, 72, and 96 hpi (Figure 4A). In the sups, at 24 hpi, the levels of CAV DNA were 7.81 ± 0.01 and 7.76 ± 0.02 log_10_ copies/mL within CAV-infected and CAV-infected + NA cells, respectively. Virus copy numbers in the sup at 48 hpi were 9.01 ± 0.1 and 8.97 ± 0.05 in the CAV-infected and CAV-infected + NA cells, respectively, and remained at such high levels at 72 and 96 hpi (Figure 4B).

CAV DNA was detected at high levels during passage until P14 in both the cells and sup of CAV-infected + NA cells (Figure 5A,B). There was no significant difference in the amount of CAV DNA between different passages in the infected or CAV-infected + NA cells (Figure 5A). Amounts of CAV DNA in the sup were 7.66 ± 0.07 log_10_ copies/mL at P1, 9.01 ± 0.02 log_10_ copies/mL at P3, and remained at these elevated levels throughout the passage (Figure 5B).

### 2.4. Recovery of CAV during Passage

CAV was successfully recovered from the CAV-infected + NA cells during the passages. Viral titers were described using the 50% tissue culture infective dose (TCID_50_). Viral titers from the cells at P1, P3, P5, P10, and P14 were 5.32, 5.32, 6.04, 5.90, and 6.26 TCID_50_/10^6^ cells, respectively. However, the virus could not be isolated from the sups at any passage (Table 1).

In order to confirm the role of NA during the passage, a subset of cells from the CAV-infected + NA line was passaged without NA (CAV-infected + NA/P7(-)NA) at P7. Viability subsequently decreased to 46.03% at P10 for these cells, which was significantly lower than that of the CAV-infected + NA at P10 (90.67%) (*p* < 0.01) (Figure 6A).

The sups of CAV-infected + NA/P7(-)NA and CAV-infected + NA cells were collected at P8, P9, and P10 and were subjected to virus isolation in MSB1 cells. The results indicated that virus titers increased to 6.77 log_10_ TCID_50_/mL in the sup of the CAV-infected + NA/P7(-)NA, whereas no virus recovery was detected from the CAV-infected + NA (Figure 6B) cells, similarly to the results presented in Table 1.

### 2.5. Genetic Analyses of Virus Recovered from Passaged Cells

CAV recovered from the CAV-infected + NA cells at P14 was designated as CAV/PI/P14; its viral genome was sequenced and compared with that of the original A2/76 strain (GenBank accession number: AB031296.1). A comparison of the protein-coding region of the CAV genome indicated that no mutations were detected in CAV/PI/P14 as compared with that of the original A2/76 strain (data not shown).

## 3. Discussion

The current study clearly indicated that CAV was maintained within infected cells in the presence of NA without killing the host cells. To the best of our knowledge, this is the first successful establishment of persistent CAV infection in MSB1 cells in the presence of NA.

The neutralizing chicken serum used in this study was heated at 56 °C for 30 min prior to use to inactivate any biologically active substances, including cytokines, which may have affected the cells separately. Although it is still difficult to completely rule out the possibility that other components in the chicken serum could have affected our results, we believe that the NA in the serum likely played an important role in the persistent infection.

Imai et al. reported that CAV could be continuously isolated from the blood cells of infected chickens at 6 weeks of age during 4-week experimental periods even after the appearance of NA, although CAV DNA could not be detected in the plasma of chickens, even via nested PCR method [23]. These previous data may suggest the presence of persistent infection. However, it is unclear exactly how long the virus persists, which blood cells are a target for persistent infection, and how CAV can exist in susceptible blood cells in the presence of NA [23]. The in vitro model of CAV persistent infection outlined here could help to reveal possible mechanisms underlying the viral persistence in chickens.

It has been well established that viruses can spread through several routes. Type I transmission occurs from infected cells to nearby or distant uninfected cells. Other forms of transmission occur from cell-to-cell based on cell fusion and budding (Type II) or from parent to progeny cells through cell division (Type III) [24]. In the present study, the high levels of CAV DNA recovered from passage supernatants in the presence of NA suggested that CAV could persist and be transmitted in the presence of NA. However, NA could potentially block type I CAV transmission; thus, this posits the hypothesis that the virus could be transmitted by other mechanisms (e.g., type II, type III, or unknown). However, demonstrating the manner in which CAV can be shed from infected cells and transmitted to other susceptible cells in the presence of NA is difficult based upon the current data. Further studies should be performed to clarify this critical phenomenon.

Previous work has shown that virions and viral DNA are detected in the blood cells of chickens infected with CAV and exhibiting NA [23], implying productive infection in the infected cells. In the present study, infectious viruses, CAV DNA, and all VPs—including capsid VP1 proteins—were detected in infected MSB1 cells in the presence of NA. These results suggested that the persistent CAV infection was productive in the presence of NA and that progeny CAV particles could be produced within the infected cells.

A genetic mutation has been considered to be a contributor to persistently infectious viral disease. The CAV VPs 1, 2, and 3 have been shown to play critical roles during viral replication [25], although there remain no reports concerning the roles of the non-protein-coding genetic region upon viral replication. In the current study, no nucleotide mutations were observed in the protein-coding region of the CAV genome within the virus recovered from persistently infected MSB1 cells, indicating that the mechanisms of CAV persistent infection are not related to molecular changes in the viral genome.

In conclusion, a model of persistent CAV infection was established in MSB1 cells in the presence of NA. CAV-infected cells remained viable during repeated passages with NA up to the fourteenth passage. CAV antigens, as well as the viral genome, were found in the CAV-infected cells cultured with NA using fluorescent antibody tests and real-time PCR assays, respectively. The mechanism(s) underlying persistent CAV infection in the presence of NA can be further elucidated with the use of this in vitro model in future studies.

## 4. Materials and Methods

### 4.1. Cell Culture, Virus, Titration, and Virus Isolation

The lymphoblastoid T cell line—MSB1, established from Marek’s disease splenic tumor in chickens [26]—were cultured in growth medium (GM) consisting of RPMI-1640 (Nissui Pharmaceutical, Tokyo, Japan) supplemented with 10% of both fetal bovine serum and Daigo’s GF21 (Wako Junyaku, Osaka, Japan) in a humidified incubator with 5% CO_2_ at 39.5 °C.

The A2/76 CAV strain isolated from the infected chickens in Japan [27] was used throughout the study. Viral titers were determined in MSB1 cells, as previously described [28]. In brief, ten-fold serial dilutions of virus suspension were cultured in 200 µL GM containing 2 × 10^5^ cells/mL; dilutions were performed in four wells of a 96-well plate. The inoculated cells were passaged at three-day intervals by transferring 40 µL of the cell suspension to a new well containing 200 µL GM (1:5 dilution). CAV-negative and -positive wells were determined after seven passages. The cytopathic effect was observed under a microscope; cell death resulting in the culture medium, causing changes in color to red, was regarded as CAV-positive [29]. Virus titers were quantified by the Behrens–Kärber method [30] and were indicated as TCID_50_.

Recovery of CAV from the in vitro infection model described below was performed, as previously reported [28], with slight modification. Briefly, a total of 2 × 10^5^ MSB1 cells in 1 mL GM were mixed with 100 µL of the CAV-infected cell suspension or phosphate-buffered saline (PBS) as a control. This treatment was performed in two wells of a 24-well plate for each cell suspension. The cells were then passaged every three days via 1:5 dilution; CAV-negative/positive wells were determined after seven passages.

### 4.2. Antiserum, Monoclonal Antibodies, and Anti-Peptide Antibodies

Chicken anti-A2/76 produced in SPF chickens was kindly provided by the National Institute of Animal Health (Tsukuba, Ibaraki, Japan). Chicken anti-A2/76 was treated at 56 °C for 30 min. The virus neutralization titer of anti-A2/76 serum was determined to be 1:1000. Neutralizing monoclonal antibodies (mAbs) used against CAV VP1 included: MoCAV/F2, MoCAV/F8, and MoCAV/F11 [3]. Anti-VP1, VP2, and VP3 peptide antibodies were produced in rabbits by Qiagen (Osaka, Japan) using the specific peptides CWDVNWANSTMYWESQ, MHGNGGQPAAGGSESC, and MNALQEDTPPGPSTC, respectively. SPF chicken serum was kindly provided by the Advanced Technology Development Center of Kyoritsu Seiyaku (Tsukuba, Japan).

### 4.3. Detection of Cell Viability

The viability of MSB1 cells was analyzed via acridine orange/propidium iodide (AO/PI) (Logos Biosystems, Gyunggi-Do, Korea), according to the manufacturer’s instructions. In brief, cell suspensions were stained with AO/PI and analyzed by a Luna-FL^TM^ fluorescence cell counter (Logos Biosystems).

### 4.4. DNA Extraction and Real-Time PCR

Viral genomes were extracted from infected cells and cell supernatants using the High Pure PCR Template Preparation Kit (Roche Diagnostics GmbH, Mannheim, Germany). Real-time PCR was conducted to detect the CAV VP1 gene, as previously described [31,32].

### 4.5. IFAT

CAV antigens were detected via IFAT, as described by Yuasa et al. [33]. Briefly, MSB1 cells were smeared on a microscope glass slide and fixed with cold acetone. The slides were incubated with chicken anti-A2/76 (1:40), MoCAV/F2, MoCAV/F8, and MoCAV/F11 (3 ng/µL), and anti-VP1, VP2, and VP3 peptide rabbit sera (1:200) for 30 min at room temperature. After washing several times with PBS, slides were incubated with rabbit FITC-conjugated anti-chicken IgG, FITC-conjugated anti-mouse IgG, or FITC-conjugated anti-rabbit IgG (Rockland, Gilbertsville, PA, USA). The cell nuclei were stained by 4, 6-diamidino-2-phenylindole (DAPI) (Sigma-Aldrich, Tokyo, Japan) for 30 min at room temperature, followed by washing with PBS three times. A fluorescence microscope (BioRevo BZ-9000, Keyence, Osaka, Japan) was used to observe viral antigens.

### 4.6. Experimental Design for In Vitro Persistent Infection

For the time-course study, a total of 1 × 10^7^ MSB1 cells were prepared and washed twice with PBS. Cells were suspended in 1 mL GM containing 1 × 10^7^ TCID_50_ CAV in a 24-well plate, then incubated in a humidified incubator with 5% CO_2_ at 37 °C for 90 min. After removing free CAV virus by washing with PBS three times, cells were seeded at 1 × 10^6^ cells/mL in GM alone (CAV-infected group, n = 3) or GM supplemented with 1:250 dilution of chicken anti-A2/76 serum (CAV-infected + NA, n = 3) or SPF chicken serum (CAV-infected + SPFs, n = 3). Uninfected cells cultured in GM served as a control (non-infected, n = 3). After culturing the cells for 24, 48, 72, and 96 h, cells and sups were collected from the CAV-infected, CAV-infected + NA, and non-infected groups. CAV-infected + NA cells were washed with PBS three times to remove NA and then resuspended in PBS for future use. Cell viability, CAV antigens, and viral DNA were analyzed by AO/PI staining, IFAT, and real-time PCR, respectively.

During the passage, MSB1 cells were infected with CAV, as described above. After washing with PBS three times, cells were seeded in a 24-well place at 4 × 10^5^ cells/mL in GM (CAV-infected, n = 3) or in GM with a 1:250 dilution of NA (CAV-infected + NA) (n = 3). Non-infected cells served as a control. Cells were harvested every 48 h, washed with PBS twice, and seeded at 4 × 10^5^ cells/mL in the corresponding media. At each passage, cell viability, CAV antigens, and CAV DNA were examined. In addition, virus isolation was performed to recover CAV from the passaged cells; for recovery of CAV from CAV-infected + NA, the cells were washed three times with PBS, then resuspended in PBS to obtain cell suspension, followed by three freeze/thaw cycles. Resultant cells and sups were subjected to virus isolation from MSB1 cells.

At P7 in the passage study, the CAV-infected + NA cell culture was divided into two groups: either with the continuous passage in the presence of NA, or passage in the absence of NA (CAV-infected + NA/P7(-)NA). Cell viability and recovery of CAV in the sups were examined.

### 4.7. Nucleotide Sequencing and Genetic Analyses

Sanger sequencing was used to determine the protein-coding sequence of the CAV genome. Two pairs of primers—CAV-CQ1F/CAV-CQ1R and CAV-CQ2F/CAV-CQ2R [34]—were used to amplify two PCR products of 1778 and 831 bp in size, respectively, containing the protein-coding region of the CAV genome. The GeneClean^®^ II Kit (MP Biomedicals, Santa Ana, CA, USA) was used to purify the PCR products after separation on agarose gels. The BigDye Terminator v3.1 Cycle Sequencing Kit (Life Technologies, Carlsbad, CA, USA) was used to perform nucleotide sequencing on an Applied Biosystems 3500 Genetic Analyzer (Life Technologies). The deduced nucleotide and amino acid sequence data were aligned and then analyzed using a ClustalW multiple alignment tool [35].

### 4.8. Statistical Analysis

The student’s *t*-test was used to identify significant differences in cell viability between the groups (CAV-infected + NA vs. CAV-infected, CAV-infected + SPFs vs. CAV-infected, or CAV-infected + NA/P7(-)NA vs. CAV-infected + NA). *p*-values < 0.05 were considered statistically significant. Data were expressed as mean ± standard deviation (S.D).

## Figures and Tables

**Figure 1 pathogens-09-00842-f001:**
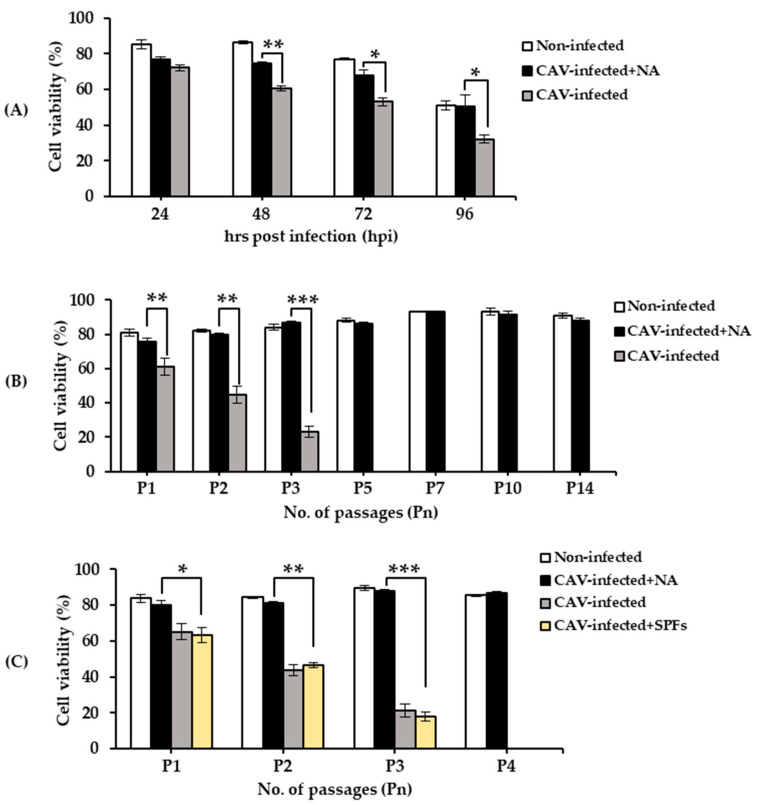
Detection of cell viability within non-infected, CAV-infected cells cultured in the absence of NA (CAV-infected), and CAV-infected cells cultured in the presence of NA (CAV-infected + NA) in (**A**) the time course and (**B**) passage. The passage study including the control group (CAV-infected + SPFs) was repeated for P1–P4 (**C**). *, **, and *** indicate significant differences (*p* < 0.05, *p* < 0.01, and *p* < 0.001, respectively). Data are represented as the mean ± standard deviation (S.D.); n = 3. CAV, chicken anemia virus; NA, neutralizing antibody; SPF, specific-pathogen-free.

**Figure 2 pathogens-09-00842-f002:**
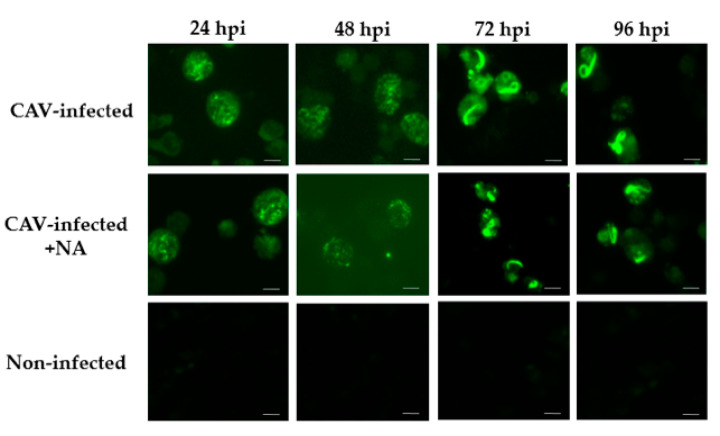
Detection of CAV antigens within CAV-infected and CAV-infected + NA during the time course. IFAT (indirect fluorescent antibody test) was conducted to capture viral antigens using chicken anti-A2/76 serum and FITC-conjugated rabbit anti-chicken IgG. CAV-infected and CAV-infected + NA were collected at 24, 48, 72, and 96 h post-infection (hpi) and used as antigens. Non-infected cells were used as negative control. Bar: 10 µm.

**Figure 3 pathogens-09-00842-f003:**
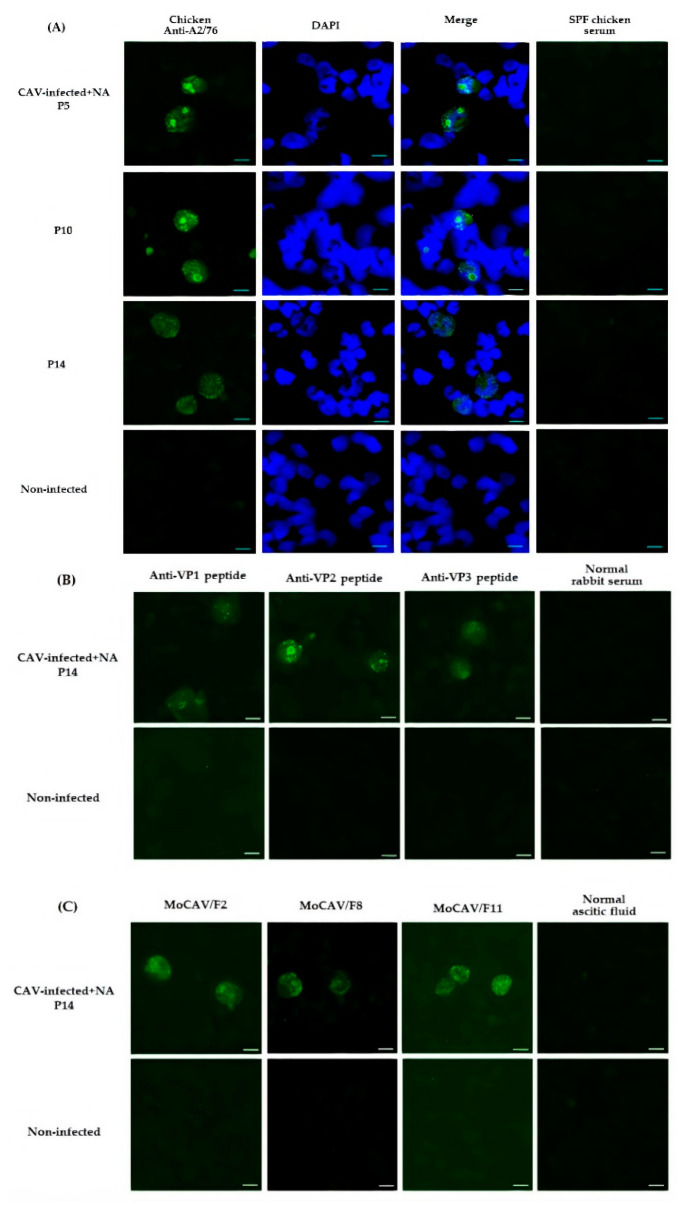
Detection of CAV antigens within CAV-infected and CAV-infected + NA cells during the passage. IFAT was conducted to capture CAV antigens using (**A**) chicken anti-A2/76, (**B**) anti-VP1, VP2, and VP3 rabbit peptide, and (**C**) MoCAV-F2, F8, and F11 sera. FITC-conjugated rabbit anti-chicken IgG or FITC-conjugated rabbit anti-mouse IgG were used as secondary antibodies. CAV-infected and CAV-infected + NA cells were collected at each passage and used as antigens. Non-infected cells were used as negative control. Bar: 10 µm.

**Figure 4 pathogens-09-00842-f004:**
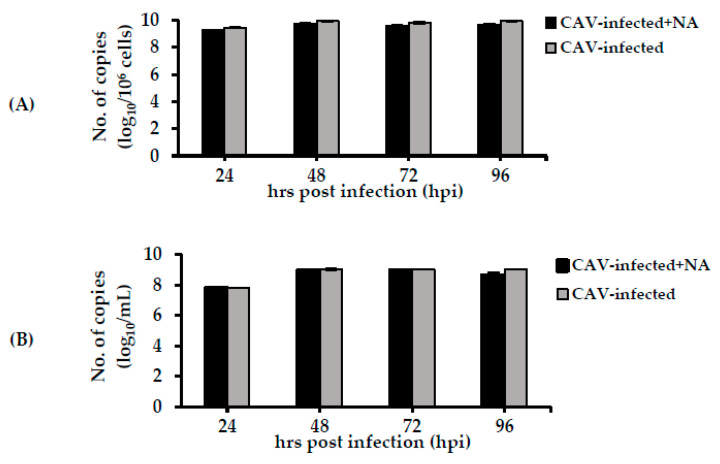
Detection of CAV DNA in (**A**) cells and (**B**) cell supernatants (sups) of the CAV-infected and CAV-infected + NA cells during the time course. Levels of viral DNA in the infected cells were determined as log_10_ number of copies/one million cells, while log_10_ number of copies/mL was regarded as levels of viral DNA in the sups. Data are shown as mean ± S.D.

**Figure 5 pathogens-09-00842-f005:**
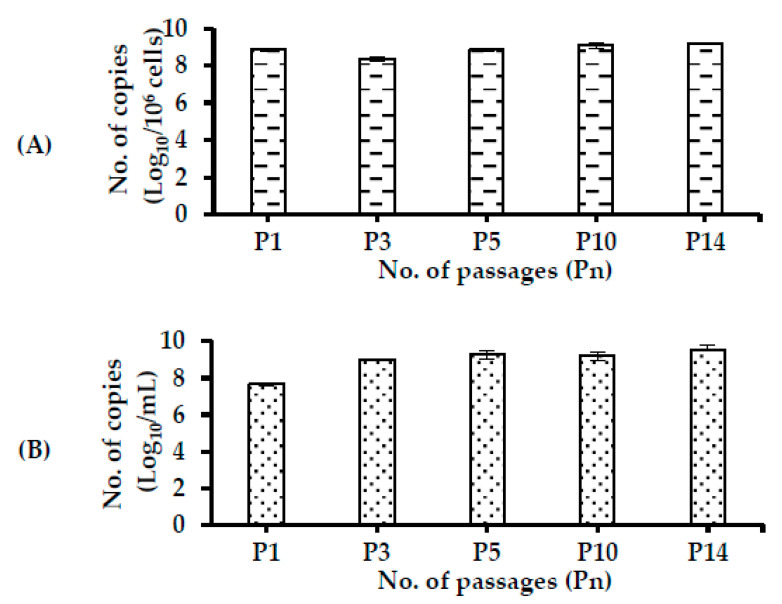
Detection of CAV DNA in (**A**) cells and (**B**) sups in the CAV-infected + NA in the passage study. Levels of viral DNA in the infected cells were determined as log_10_ number of copies/one million cells, while log_10_ number of copies/mL was regarded as levels of CAV DNA in the sups. Data are shown as mean ± S.D.

**Figure 6 pathogens-09-00842-f006:**
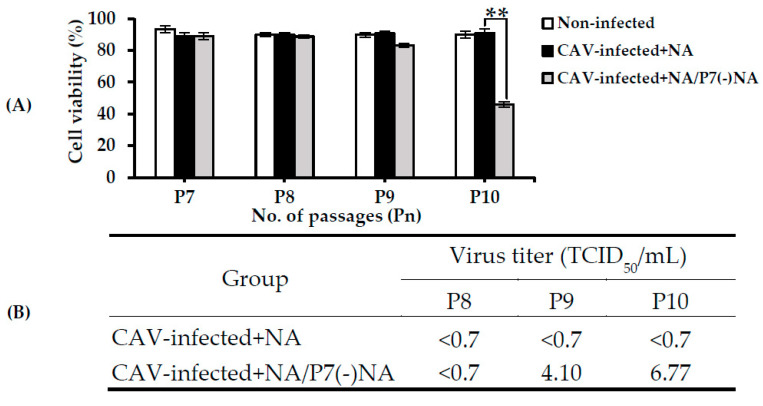
(**A**) Detection of cell viability and (**B**) recovery of CAV in the sup of CAV-infected + NA/P7(-)NA in the passage study after removing NA. ** indicates a significant difference (*p* < 0.01). Data are shown as mean ± S.D. NA/P7(-)NA, NA line passaged without NA at P7.

**Table 1 pathogens-09-00842-t001:** Recovery of CAV from CAV-infected + NA cells during the passage.

Sample	CAV Isolation (TCID_50_/10^6^ cells)			
	No. of Passages			
	P1	P5	P10	P14
cells	5.32	6.04	5.9	6.28
sup	<0.7	<0.7	<0.7	<0.7

CAV isolation was conducted from the MSB1 cells persistently infected with CAV in the presence of NA (CAV-infected + NA cells). CAV, chicken anemia virus; TCID_50_, the 50% tissue culture infective dose; sup, supernatant.

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
