# Peer review of "Establishment of an In Vitro Model of Persistent Chicken Anemia Virus Infection"

_pathogens, 2020, doi:10.3390/pathogens9100842_

Round 1
Reviewer 1 Report
This paper describes the successful establishment of an in vitro model of persistent Chicken Anemia Virus infection. CAV infections are a recurring problem in the chicken industry, especially in flock kept for production of eggs for vaccine production. A model for persistent infections can help in developing strategies to control this.
The paper is well written, but I have some comments on to the experimental design and conclusions drawn.
The use of a neutralizing chicken serum against A2/76 clearly has an effect on culture viability. Key question is, if the antibodies play a role in this, and if so, how. The authors do not mention the possibility that other factors in serum than antibodies may also affect cell viability and it is my belief that controls for that are lacking.
- What is the effect of culturing in a 1:250 dilution of a non-neutralizing chicken serum? Could it be that cytokines or other chicken constituents have an effect
- MoCAV monoclonals are suggested to be neuralizing as well: do these have the same effect as chicken serum A2/76?
If the virus remains intracellular, how could antibodies help in maintaining cell viability?
The abstract states that the aim was to clarify the mechanism, I do not see sufficient data to support that statement
I agree that a model for persistent CAV infections has been set up, but for publication in Pathogens additional control experiments are necessary.
Reviewer 2 Report
This study describes the creation and validation of an in vitro model of neutralizing antibody dependent persistence of Chicken Anemia Virus in DSCC-MSB1 cell culture.
Figure 2: As there are differences in cell viability at these time points, adding panels showing DAPI staining of these cells would highlight cell viability and show nuclear condensation in apoptic cells.
Figure 3: The resolution of the figure is relatively low, and the panels are quite small which makes it difficult to see some of the finer details of virus localization. Additionally, in panel 3A, the DAPI staining does not appear to correspond to the cells stained with Chicken Anti-A2/76. Adding a 4th column of images with the merge of the DAPI and Chicken Anti-A2/76 staining would allow better visualization of the cellular location of viral staining.
Figure 5: The manuscript text and figure legend both reference this data including both CAV-infected and CAV-infected+NA cells, however, the figure appears to only depict the CAV-infected+NA cells. Please clarify or add in the missing CAV-infected data.
Table 1: It would be interesting to see the comparative titer from CAV-infected cells without NA in this table.
Table 1 and Figure 6: Please comment on the limit of detection for the ND samples in the corresponding legends.
Section 2.5: Were these sequences also identical to a published reference sequence? If so, please provide the accession number. If not, please deposit the consensus sequence in an appropriate database and include the accession number in the manuscript.
Section 3, sentence 2: Should “full” be “first”?
Although it may be outside the scope of this study, it would be interesting to at least comment or speculate on the respective levels of apoptosis observed between CAV-infected and CAV-infected+NA during passaging. It would improve the discussion on Type I transmission to include a bit of background on budding and release of CAV from infected cells, along with the role of apoptosis in the release of virions, if this has been described previously.
Reviewer 3 Report
The authors performed research aimed to establish an in vitro model to reproduce and try to clarify the reasons behind the persistent infection of host cells with the Chicken Anemia Virus (CAV). The CAV infection of eggs from SPF chickens, used as a substrate for poultry and human vaccines production has been a problem in the past, and CAV infection is difficult to eradicate in a flock once entered. In this study an in vitro model based on CAV-infected MSB1 cells adding neutralizing antibodies to the media to simulate the persistent infection in chicken cells and analyzing the survival of cells and virus throughout subsequent passages. The results showed that CAV is able to survive in CAV-infected cells cultured in the presence of neutralizing antibodies without affecting host cell viability.
The manuscript is very well written, quite short but well-focused, and well-designed. The methods are clearly described, and the Results are appropriately discussed in the context of the literature. I recommend the publication.
I just request a small clarification regarding the materials and methods section:
- The authors supplemented the culture medium with 1:250 dilution of chicken anti-A2/76 serum. Why did they choose 1:250 dilution of neutralizing antibodies? Is there a previous work in which this dilution was tested?
Round 2
Reviewer 1 Report
First, I would like to thank the authors for the answers to my questions and for addressing these issues in the manuscript where possible. I realize that it is not possible to address some of the questions experimentally on a short term. The authors are aware that the model needs to be further tested, but in principle the model is a valuable tool to study CAV persistence. There is still work to be performed to find an explanation for persistent CAV infections, an important issue for poultry vaccine production.
Author Response
We thank the Reviewer for providing constructive comments to improve the quality of our manuscript. We performed additional experiments to confirm the effect of non-neutralizing chicken serum (SPF chicken serum), and confirmed that presence of non-neutralizing chicken serum does not suppress the cell death of CAV-infected cells. The changes made in the revised manuscript are highlighted in red font.